# Application of Bioelectrical Impedance Analysis (BIA) to Assess Carcass Composition and Nutrient Retention in Rabbits from 25 to 77 Days of Age

**DOI:** 10.3390/ani12212926

**Published:** 2022-10-25

**Authors:** Alejandro Saiz del Barrio, Ana Isabel García-Ruiz, Joaquín Fuentes-Pila, Nuria Nicodemus

**Affiliations:** 1Trouw Nutrition R&D Poultry Research Centre, 45950 Casarrubios del Monte, Spain; 2Departamento de Economía Agraria, Estadística y Gestión de Empresas, 28040 Madrid, Spain; 3Departamento de Producción Agraria, ETSI Agrónomica, Alimentaria y de Biosistemas, 28040 Madrid, Spain

**Keywords:** chemical composition, growing rabbits, in vivo method, bioelectrical impedance, prediction equations, nutrient retention

## Abstract

**Simple Summary:**

The determination of carcass composition in growing rabbits is of utmost importance for research, technical and commercial purposes, to optimize nutrient retention and carcass quality. Additionally, a non-invasive method is desired for animal welfare protection purposes. In this study, it is shown that the bioelectrical impedance method is an accurate non-invasive method to determine growing rabbits’ carcass composition.

**Abstract:**

The aim of this study was to assess and validate, using independent data, the prediction equations obtained to estimate in vivo carcass composition using bioelectrical impedance analysis (BIA) to determine the nutrient retention and overall energy and nitrogen retention efficiencies of growing rabbits. Seventy-five rabbits grouped into five different ages (25, 35, 49, 63 and 77 days) were used in the study. A four-terminal body-composition analyzer was applied to obtain resistance (Rs, Ω) and reactance (Xc, Ω) values. All the animals were stunned and bled at each selected age, and the chilled carcasses were analyzed to determine water, fat, crude protein (CP), ash and gross energy (GE). Multiple linear regression analysis was conducted to determine the equations, using body weight, length and impedance data as independent variables. The coefficients of determination (R^2^) to estimate the content of water, protein, fat and ash in grams, and energy in Mega Jules(MJ), were: 0.99, 0.99, 0.95, 0.96 and 0.98, respectively, and the relative mean prediction errors (RMPE) were: 4.20, 5.48, 21.9, 9.10 and 6.77%, respectively. Carcass yield (%) estimation had values of 0.50 and 10.0 for R^2^ and RMPE, respectively. When water content was expressed as a percentage, the R^2^ and RMPE were 0.79 and 1.62%, respectively. When the protein, fat and ash were expressed as a percentage of dry matter (%DM) and the energy content as kJ/100 g DM, the R^2^ values were 0.68, 0.76, 0.66 and 0.82, respectively, and the RMPEs were 3.22, 10.5, 5.82 and 2.54%, respectively. Energy Retention Efficiency was 20.4 ± 7.29%, 21.0 ± 4.18% and 20.8 ± 2.79% from 35 to 49, from 49 to 63 and from 35 to 63 d, respectively. Nitrogen Retention Efficiency was 46.9 ± 11.7%, 34.5 ± 7.32% and 39.1 ± 3.23% for the same periods. Energy was retained in body tissues for growth with an efficiency of approximately 52.5%, and the energy efficiency for protein and fat retention was 33.3 and 69.9%, respectively. This work shows that BIA is a non-invasive and good method to estimate in vivo carcass composition and to determine the nutrient retention of growing rabbits from 25 to 77 days of age.

## 1. Introduction

Several authors have analyzed the chemical composition of rabbit carcasses at the end of the growing period [1,2,3,4] at different slaughter weights [5,6,7], or even in suckling rabbits, from 0 to 35 d of age [8].

Traditionally, the reference method used to determine carcass composition and nutrient retention has been comparative slaughter. However, this method is invasive and expensive and does not allow researchers to follow the evolution of the animal’s chemical composition over time, as the retention of nutrients is calculated using different animals at the beginning and end of the study period. However, the evolution of composition with the age of the carcass using the same animal to determine nutrient retention has not been studied previously.

The in vivo determination of carcass composition during the growing period could be interesting to study the variation in carcass composition with aging and to determine nutrient retention without slaughtering the animals. Some authors have estimated body composition using in vivo methods in rabbits such as Magnetic Resonance (MRI) [9,10,11,12], and the method of dilution with deuterium oxide (D2O) in rabbit does [13,14]. Milisits et al. [14] also used the Total Body Electrical-Conductivity method (TOBEC) in growing rabbits.

Bioelectrical impedance (BIA) is another in vivo method based on the resistivity of the body when an alternating electrical current is introduced through the animal. Estimation of the chemical carcass composition of animals using the BIA method was successfully used in swine [15,16], lambs [17], beef [18] and broiler chickens [19,20]. Recently, some authors also observed that this technique could be a practical, easy and quick method to predict the whole-body composition of rabbit does [21,22] and growing rabbits [23].

The aim of this study was to determine and validate, using independent data, the prediction equations developed using the BIA method to estimate in vivo carcass composition and assess its application, to determine nutrient retention in a group of rabbits aged from 25 to 77 days. 

## 2. Materials and Methods

The present study was approved by the Trouw Nutrition R&D Animal Care Ethical Committee, reviewed by an external Animal Care Committee (Hospital de Ciudad Real) and authorized by the Livestock Service of Castilla-La Mancha, Spain (Animal Usage protocol: 12-2021). Rabbits were managed according to the Spanish Regulations of Usage of Animals in Research (Royal Decree 53/2013) [24] in agreement with the European Parliament (2010).

### 2.1. Animals and Housing

This study was carried out at the Poultry Research Centre of Trouw Nutrition, located in Casarrubios Del Monte (Toledo, Spain), in collaboration with the Universidad Politécnica de Madrid. One hundred New Zealand ×Californian rabbits were selected at random and blocked by litter (75 animals for the calibration group and 25 for the external validation group). Animals were allocated with does in polyvalent cages measuring 380 mm × 1000 mm × 320 mm until weaning (25 or 35 d, depending on the slaughter age). Then, animals were randomly housed individually in flat-deck cages measuring 250 mm × 440 mm × 300 mm. Rabbits were kept under controlled environmental conditions (room temperature between 16 and 24 °C, with a light:dark cycle of 16:8 h; the light was switched on at 07:30 h).

### 2.2. Diets

The composition of diets is shown in Table 1. Animals were fed a commercial diet for unweaned rabbits, Cunilactal Super, until weaning at 35 d, and a standard-growing rabbit diet from 35 d until slaughter. Both diets were formulated to meet animal requirements according to De Blas and Mateos [25]. Feed and water were supplied ad libitum during the whole trial. Animals were not supplied with any antibiotics during the growing period.

### 2.3. Bioelectrical Impedance Analysis Measurements

Bioelectrical impedance was measured at different ages (suckling rabbits at 25 and 35 d, and growing rabbits at 49, 63 and 77 days of age) to cover growing periods and slaughter ages commonly applied in commercial rabbit farms. In order to obtain a representative sample of animals, at each timepoint, rabbits from a wide weight range were selected. At 1100 h, the BW and feed intake of animals were registered, and a four-terminal body-composition analyzer (Model BIA-101; RJL Systems, Detroit, MI, USA) was used to register resistance (Rs, Ω) and reactance (Xc, Ω) values. Animals were placed on a non-slip and plain surface (wood or polystyrene) located above the cages to allow animals to be quiet and calm. Then, each pair of electrodes (1 black and 1 red) was inserted subcutaneously using a disposable needle (21 gauge × 1.5″ (0.8 mm × 40 mm)), at 4 cm from the base of the ears and at 4 cm from the base of the tail, respectively, as shown in Figure 1. In each animal, BIA measurement was performed twice (one every half-hour) between 11 and 13 h to determine repeatability. The distance (D) between the internal electrodes (red ones) and the body length (L) of the animals was measured. The impedance (Z) value was calculated as Z = (Rs^2^ + Xc^2^)^1/2^. The volumes (vol1 and vol2) of the animals were calculated as vol1: D^2^/Rs and vol2: D^2^/Z in order to determine the relationship between impedance and volume established by Lukaski et al. [27].

### 2.4. Carcass Composition Measurement

Fifteen animals per age (25, 35, 49, 63 and 77 days) were randomly selected to be included in the calibration carcass composition group (CG). After performing the BIA analysis, animals were stunned, slaughtered and bled. The skin, organs and digestive contents were removed, and the carcasses were chilled, weighed and stored at −20 °C. Later, animals were slowly thawed for 24 h and ground using an industrial meat chopper (Cruells, C-15 EN 60742, Spain). Three representative homogeneous samples of the ground material were collected. One sample was immediately sent to the laboratory for determination of the DM content and the other two samples were refrozen at −20 °C. Afterwards, samples were freeze-dried for 72 h and ground through a 1 mm screen. The same procedure was applied to a different group of animals (5 animals per age) considered as a validation carcass composition group (VG).

### 2.5. Digestible Energy and Protein Carcass Retention

Twenty animals were used to estimate the daily energy and nitrogen retention (ER and NR, respectively) and the ER and NR efficiency of carcasses between 35 and 63 days. Body weight and feed intake of those animals were registered, and the daily digestible N and E intake (DN and DE, respectively) and the average daily growth rate and feed conversion rate were calculated. Total carcass nitrogen, fat and energy content were calculated using two methods: the BIA prediction equations, and the values of analyzed carcass composition obtained previously. In both cases, the retention of the nutrients was determined in the periods of 35–49 d, 49–63 d and 35–63 d using the difference between the nitrogen and energy content at the beginning and at the end of each period. The overall carcass Nitrogen Retention Efficiency (NRE, %) and carcass Energy Retention Efficiency (ERE, %) were calculated as follows

NRE, % = 100 × (carcass NR/DN intake)

ERE, % = 100 × (carcass ER/DE intake)

Moreover, a factorial method was used, in which the requirements for maintenance were separated from those of production. Using this method, the energy efficiency for fat and nitrogen deposition was estimated via regression analysis between DE intake and energy retention as nitrogen and fat. The energy and nitrogen efficiency for growth was estimated via linear regression between DE intake and retained energy, and DN intake and retained nitrogen, respectively.

### 2.6. Chemical Analysis

The dry-matter content of the ground material samples was determined by mixing 5 g of sample with 20 g of sea sand and 5 mL of ethanol, and then, drying at 103 °C for 24 h, following the ISO method (1442) [28]. Those samples and diets were both analyzed following the AOAC methods (2002) [29]: DM (934.01), CP (Dumas Method, N × 6.25; 968.06), ash (942.05) and fat (RD 609/1999 nº4, previous acid hydrolysis). The GE content of the meat was determined using an adiabatic calorimetric bomb (model 6100; Parr Instrument Company, Moline, IL, USA). The DE and DN of the diets were estimated using the digestibility coefficient of each raw material described by Maertens et al. [26].

### 2.7. Statistical Analysis

The statistical program SAS/STAT (SAS Inst. Inc., Cary, NC, USA) was used for all statistical analyses. Levene’s test was conducted to test the homogeneity of variances. Linear and quadratic responses of carcass composition were studied using the PROC GLM method.

#### 2.7.1. Repeatability of BIA Measurements and Correlation between Variables

Repeatability (SR; intraseries variability of BIA measurements within rabbit) was estimated using the VARCOM procedure in SAS and was calculated as S_R_ = √(Se)^2^, in which Se was the expected variance of error. The CV of repeatability (CVR) was calculated as the relationship between SR and the mean value of the BIA measurements, expressed as a percentage. Correlation coefficients between BIA measurements and carcass composition were calculated using the procedure CORR.

#### 2.7.2. Selection of Variables and Validation of Equations

In order to select the regression models that bets explained the variation in the dependent variables, the RSQUARE option of PROC REG was used, using the chemical composition data of the CG animals. The dependent variables included were water (expressed as % and as g), CP, ash, fat (expressed as %DM and g) and energy (kJ/100 g DM and MJ). Independent variables chosen as candidates to be included in the regression model were age, BW, BW^2^, L, L^2^, D, D^2^, Rs, Rs^2^, Xc, Xc^2^, Z, Z^2^, vol1 (D^2^/Rs) and vol2 (D^2^/Z). The model was selected using Mallows’ Cp statistics [30], which should be lower than or equal to p + 1 (p is the number of independent variables included in the model) for avoiding biases because of the omission of relevant explanatory variables. When this criterion was satisfied, the model with the minimum value of the statistics—SP Statistics (SP) [31], Final Prediction Error (JP) [31,32], Amemiya’s Prediction Criteria (PC) [32,33] and Akaike’s Information Criterion (AIC) [34]—was selected. After selecting the independent variables to be included in the multiple linear regression (MLR) models, the parameter estimation was performed using PROC REG. The chemical composition data of the VG animals were used to validate the regression equations with independent data. The prediction accuracy of the equations was measured using the mean prediction error (MPE) and was calculated as the root square of the sum of the squares of the residuals between the actual values of the carcass composition parameters, estimated using chemical analysis techniques, and the predicted values, divided by the number of observations. Relative mean prediction error (RMPE, %) was calculated as the ratio between MPE and the average of the carcass composition parameters’ actual values. The difference between the observed and estimated values was obtained using prediction equations (in the validation group, it was compared using a paired t-test). PROC REG was used to obtain residuals between the analyzed and predicted carcass protein and fat content, including the confidence interval of the prediction (95%).

#### 2.7.3. Nutrient Retention Analysis

A paired *t*-test was also used to compare energy, nitrogen and carcass retention, and ERE and NRE, calculated as the difference in the total carcass energy and nitrogen content of animals using Multiple Linear Regression equations against the analyzed carcass energy and nitrogen content. The PROC REG of SAS was used to stablish different linear regressions between DE intake and energy retention; average daily growth (ADG) and energy retained as protein (P) and fat (F); and DN intake and nitrogen retention in the carcass.

## 3. Results

### 3.1. Impedance Measurements and Repeatability

The mean values, range and variability of Rs, Xc, Z, D, L and BW per age of the CG animals are shown in Appendix A. When the age of the animals increased, the values of Rs and Z decreased, with values of 121 Ω and 122 Ω, respectively, at 25 d, and values of 63.0 Ω and 65.0 Ω, respectively, at 77 d. The values of Xc increased from 25 (16.4 Ω) until 63 d (24.5 Ω), and then, decreased at 77 d (16 Ω).

In Appendix A the values of these parameters of the VG animals are shown. The values were similar to those obtained from the CG animals, and likewise, when the age of the animals increased, the values of Rs and Z decreased, with values of 130 Ω and 132 Ω, respectively, at 25 d, and values of 75.2 Ω and 76.7 Ω, respectively, at 77 d. Xc increased with age, from 18.5 Ω to 24.3 Ω at 25 and 63 d, respectively, but decreased to 16.2 Ω at 77 d.

In Table 2, the values of repeatability (SR, Ω) and the coefficient of variation of repeatability (CV_R_, %) of Rs and Xc are shown. The value of repeatability was higher for Rs (15.3 Ω) than for Xc (3.44 Ω), whereas the values of CV_R_ were similar (Rs = 15.9; Xc = 17.6).

The animals’ average daily feed intake, average daily gain, and feed conversion rate per period are shown in Table 3. Fifteen animals per age were slaughtered to analyze carcass composition at each age; therefore, the number of controlled animals decreased.

### 3.2. Carcass Composition

The results of the BW, carcass weight (CW), carcass yield (CY) and the analyzed carcass composition of the CG animals are shown in Table 4. The BW was increased in a quadratic way (*p* < 0.001) with values from 370 to 2930 g at 25 and 77 d, respectively. Carcass weight was increased in a linear (228 g at 25 days to 1838 g at 77 days; *p* < 0.001) and quadratic way (*p* < 0.001). Carcass yield followed a linear (*p* < 0.001) and a quadratic (*p* < 0.001) evolution with a maximum value at 25 days (64.3%) and a minimum value at 35 d (45.7%), increasing later to 62.8 at 77 days.

The carcass composition also changed with age. The water content tended to decrease linearly with age (*p* < 0.10; from 71.6 to 64.1%). Protein and ash content, expressed as %DM, also decreased linearly (*p* < 0.001) with age, with values between 59.7 and 49.4%DM, and between 16.8 and 11.5 %DM, respectively. A maximum value for carcass water, protein and ash content was found at 35 days (72.1%, (*p* < 0.001); 61.5%DM, (*p* < 0.001) and 18.5%DM, (*p* < 0.005), respectively). Fat content varied in a quadratic way with age (*p* = 0.005), with values from 19.6 to 34.3%DM at 25 and 77 d, respectively, and a minimum value (19.0%DM) at 35 d. Energy content increased linearly with age (*p* = 0.006), with values between 2054 and 2557 kJ/100 g DM at 25 and 77 d, respectively. The animals also reached a minimum energy value of 1980 kJ/100 g DM at 35 d (quadratic *p* < 0.001). When the carcass composition was expressed in grams and the energy in MJ, all parameters increased with age in a linear (*p* < 0.001) and quadratic way (*p* < 0.001).

### 3.3. Correlation between Variables

The results of the correlation analysis between variables are shown in Appendix A. Reactance was positively correlated with fat content (r = 0.24, *p* < 0.05), while resistance was positively correlated with carcass water, ash and protein content (r = 0.55, *p* < 0.001; r = 0.54, *p* < 0.001; and r = 0.40, *p* < 0.005, respectively) and negatively correlated with fat and energy (r = −0.44, *p* < 0.001 and r = −0.55, *p* < 0.001, respectively). CY was negatively correlated with protein and fat content (r = −0.54, *p* < 0.001 and r = −0.36, *p* < 0.001, respectively) but positively with water, ash and energy content (r = 0.80, *p* < 0.001; r = 0.83, *p* < 0.001 and r = 0.65, *p* < 0.001, respectively). Moreover, age was negatively correlated with water, ash and protein content (r = −0.97, *p* < 0.001; r = −0.95, *p* < 0.001; and r = −0.89, *p* < 0.001, respectively) and positively correlated with fat, energy and carcass yield (r = 0.95, *p* < 0.001; r = 0.97, *p* < 0.001; and r = 0.84; *p* < 0.001).

### 3.4. Regression Equations

The regression equations obtained using multiple linear regression (MLR) from the CG are shown in Appendix A, with the estimate, SE and *p*-values of each variable, and the R^2^, residual SD, CV, Cp and probability of the model of each equation.

### 3.5. Validation of Prediction Equations

The comparative results obtained from the validation with independent data, determined using the MLR equations, are shown in Table 5. The adjustment of the prediction equations obtained using MLR for the animal carcass composition was higher (R^2^ > 0.90) when it was expressed in grams than when it was expressed as a percentage. The RMPE of all equations was higher when the chemical composition was expressed in grams than when it was expressed as a percentage.

In Table 6, the results of the pairwise t-test conducted to compare the analyzed and the predicted values of animal carcass composition, obtained via MLR, are presented. No significant differences between the analyzed and predicted values of carcass composition were found, except in the case of ash content expressed in grams, which showed a tendency to be underestimated (*p* < 0.10) (analyzed: 36.6 ± 27.9 g; predicted by MLR: 34.3 ± 23.9 g).

The distribution of the residuals obtained via differences between the analyzed and predicted values using the MLR equations, expressed as a DM percentage and in grams, are shown in Figure 2. For protein and fat, when composition was expressed in grams, the residual distribution was less homogenous at higher content values. In the case of DM percentage equations, when the protein content of a carcass was higher than 60%DM, there was an overestimation in the prediction of protein, and an underestimation when the protein content was lower than 50%DM. On the contrary, when the fat content was higher than 30%DM, there was an underestimation, and below 20%DM, the equation overestimated the carcass fat content.

### 3.6. Energy and Nitrogen Carcass Retention

The values of DE and DN intake, the comparison between the protein, fat and energy retained, and the overall and NRE are shown in Table 7. DE intake was 196, 340 and 268 Kcal/d in the periods of 35–49 d, 49–63 d and 35–63 d, respectively. Nitrogen intake was 1.44, 2.57 and 1.97 g/d for the same periods, respectively. The average daily growth rate values were 41.1, 51.1 and 45.5 g/d in the periods of 35–49 d, 49–63 d and 35–63 d, respectively.

The values of analyzed energy, nitrogen and fat retained were, respectively, 35.8 Kcal/d, 0.654 g/d and 2.73 g/d between 35 and 49 d; 75.6 Kcal/d, 0.896 g/d and 4.51 g/d between 49 and 63 d; and 55.7 Kcal/d, 0.770 g/d and 3.62 g/d between 35 and 63 d. The values of energy, nitrogen and fat retained estimated by the MLR equations were, respectively, 40.1 Kcal/d, 0.675 g/d and 1.93 g/d between 35 and 49 d; 71.4 Kcal/d, 0.867 g/d and 4.33 g/d between 49 and 63 d; and 55.8 Kcal/d, 0.771 g/d and 3.13 g/d between 35 and 63 d. No differences (*p* > 0.05) were found between the analyzed and estimated nutrient retention.

When the global retention efficiency was calculated using analyzed nitrogen and energy, the values of ERE were: 18.2%, 22.2% and 20.7% at 35–49 d, 49–63 d and 35–63 d, respectively. The NRE values were: 45.4%, 35.7% and 39.1% at 35–49 d, 49–63 d and 35–63 d, respectively. These values were the same (*p* > 0.05) as those calculated using the predicted nitrogen and energy values using MLR equations, with ERE values of 20.4%, 21.0% and 20.8% at 35–49 d, 49–63 d and 35–63 d, respectively, and NRE values of 46.9%, 34.5% and 39.1% at 35–49 d, 49–63 d and 35–63 d, respectively.

Values of R^2^ around 0.45 and 0.81 for ERE and NRE, respectively, were obtained when the overall analyzed and predicted retention efficiency values were plotted and compared (Figure 3).

## 4. Discussion

### 4.1. Impedance Measurements and Repeatability

The impedance measurement is clearly affected by the chemical carcass composition. Fat is electrically resistant, while minerals contained in the water, ashes and protein are conductors. The values of resistance and reactance obtained in this study were higher than the values obtained by Saiz et al. (2017) [23] in heavier animals, but also higher than the values of pigs [15] or lambs [35]. These findings can be explained by differences in the volume of animals, described by Lukaski et al. [27] as: Volume = √ (Length^2^/Impedance).

The values of CVR were similar to those obtained by Saiz et al. [23], but still higher than in humans (between 0.3 and 2.8%) [27,36,37,38]. As was mentioned by Saiz et al. [23] this could be due to the differences in the way electrodes are applied. In humans, (the electrodes are applied using a patch on the skin, while in rabbits, needles through the skin are used, which could lead to higher variation. That is why it is recommended, in rabbits, to take at least two measurements per animal. The measurements were performed in long intervals and the variability between the two consecutive measurements also depended on the state of the animals (feed and water intake and digestive content). Therefore, as no fasting period was planned, this variability was included in the error term. It was carried out in this way because the final aim was that this method could be applied at the field level. In standard conditions, when we perform the measurements, we cannot know whether an animal has drunk, eaten and excreted. We also tried to take the BIA measurements without modifying the natural behavior of the animals.

### 4.2. Carcass Composition Measurement

Chemical composition is expressed on a DM basis because water content varies with age and needs to be corrected to compare the changes in the chemical composition with age, without the effect of water content differences.

The carcass yield and chemical composition values were similar to those obtained by other authors [5,39,40]. The carcass yield and the fat and energy content increased with age, as also found in the whole-body chemical composition [23] with maximum levels of water, protein and ashes and minimum levels of fat and energy at 35 d of age. Between 25 and 35 d, more protein and tissue associated with a higher level of water and ash was deposited on the carcass, while fat was deposited in a higher quantity from that moment onwards. These results can be explained by the allometric growth of the young animals [41,42,43], which reflects the different growth rates of animal tissues depending on the age. The lowest carcass yield observed between 25 and 35 d could be explained by the increase in solid feed intake in this period, leading to higher growth of the intestinal tract and to a lower carcass yield. From that moment, the rabbit started depositing more fat and the carcass yield increased with age, as observed in previous studies [44].

### 4.3. Correlation between Variables

As was already mentioned, fat and energy content were highly correlated with age and the opposite was observed for protein, ash and water. These results agree with previous studies in other species [15,19,20,45], as well as in rabbits [8,22,39].

### 4.4. Validation of Prediction Equations

The lack of differences between the analyzed and predicted chemical composition values corroborates the accuracy of these equations. The RMPE values of the equations, expressed as a DM percentage, were lower than those expressed in grams, indicating higher precision in the first set of equations. However, the second ones showed higher R^2^ values, probably due to the higher variance interval. This result has also been observed by Pereda [22] and Saiz et al. [23].

Studying residual distribution, using MLR prediction equations, a slight underestimation of the carcass protein content of rabbits with the lowest protein content and an overestimation in rabbits with the highest values was observed, the reverse of what was observed by Saiz et al. [23]. On the contrary, fat is overestimated in rabbits with the lowest carcass fat content and underestimated in those with the highest values. In the work previously mentioned, only an overestimation of the predicted fat content was found at the lowest fat levels. The residuals were more homogenous when the chemical composition was expressed as a percentage because the range was reduced in comparison with when it was expressed in grams.

Comparing these results with those obtained in other works whereby other in vivo techniques were used to predict the carcass chemical composition of growing rabbits, it was detected that none of them showed values of RMPE rendering it impossible to make comparisons. Most of the works only showed the R^2^ values or CV (%). The R^2^ of fat content (g) prediction in this study was higher (0.97) than the values shown by Köver et al. [11], who found an r-value of 0.39 (R^2^ = 0.15) using the MRI method. Fekete and Brown [12] used both TOBEC and the deuterium dioxide method to estimate empty body fat. In the case of the TOBEC method, the authors observed an R^2^ value of 0.97 in animals who did not receive any food or water before slaughter. So, the adjustment of the BIA method is as good as or even better that any other technique applied. Additionally, it is relatively cheaper and simpler than the other methods used.

### 4.5. Carcass Energy and Nitrogen Retention

The energy and nitrogen retention and the efficiency of retention (ERE and NRE) in the carcasses were calculated in animals between 35 and 63 d (the most common length for a growing period in a Spanish commercial rabbit farm), using both the overall method (which considers the maintenance protein and energy requirements) and the factorial method.

In this study, the nitrogen, fat and energy retention in carcasses was calculated using the analyzed and predicted carcass composition using the MLR equations; we found no differences between both methods of carcass composition determination, which corroborates the good accuracy of the BIA method. In the same way, the overall ERE and NRE values were equal using both nitrogen and energy-content determination methods. Contrary to ERE, NRE was higher in the first growing period (from 35 to 49 days) than in the second one (from 49 until the end of the growing period). This fact can be explained by changes in the allometric growth of animals [41,42,43]), as previously mentioned. During the first part of the growing period, rabbits deposit more protein in the carcass, using some of their energy for this, while in the last part of the growing period, the animal deposit less protein in the muscle and more energy as fat. In this regard, some authors [6,7] observed higher values of ERE (around 27%) but similar values of overall NRE (39.2%) than in the present study. On the other hand, other authors [5,46,47] found similar ERE values (around 20%) but higher values of NRE (near 50%). Several authors [1,5,48] studied the nutrient retention efficiency used for growth, and fat and protein retention using the factorial method. Similar to the values of these authors, in this study, the energy was retained in the body tissues with an efficiency of approximately 50%. On the other hand, a slightly lower proportion of energy retained as protein (33.3%) was found than that obtained by these authors (40%, on average), but the DE retained as fat (69.9%) was similar to the values shown in those papers. The nitrogen retained in body tissues was 77%, higher than that obtained by Partridge et al. [1], who obtained values between 59 and 67%, and than the 56% observed by Xiccato and Trocino [49]. An explanation for this could be found in the genetic improvement in rabbits’ protein utilization over the years. On the other hand, De Blas et al. [5] observed retention of nitrogen in the body tissues of 90%, using a wide range of dietary protein levels in the Spanish giant rabbit breed. These results could lead to the conclusion that the BIA method can be considered an accurate tool to estimate energy and nitrogen retention.

## 5. Conclusions

It can be concluded that: (1) the estimation of carcass composition is more accurate with equations expressing the chemical composition as a percentage than in grams; (2) The bioelectrical impedance (BIA) method can be applied to determine rabbits’ carcass composition during the whole growing period; (3) BIA predicts, with accuracy, the nutrient retention of growing rabbits, showing values such as those obtained using the comparative slaughter technique.

## Figures and Tables

**Figure 1 animals-12-02926-f001:**
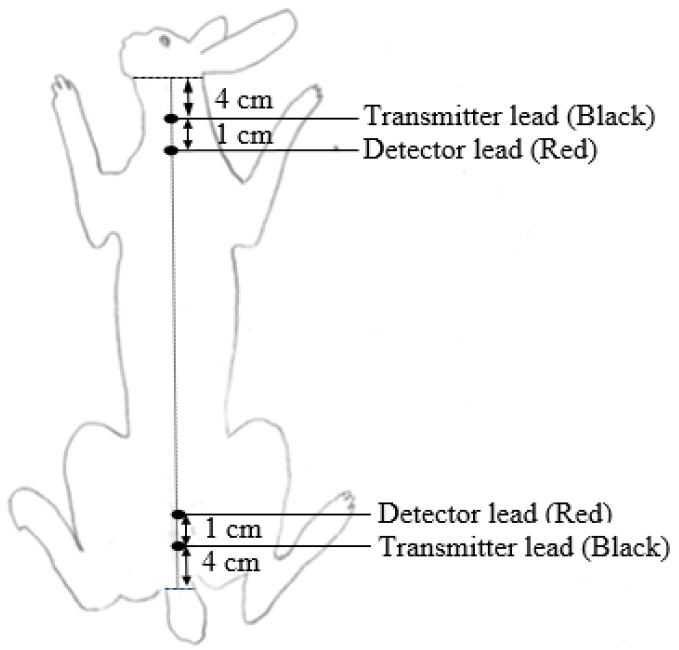
Electrode locations in live rabbit body.

**Figure 2 animals-12-02926-f002:**
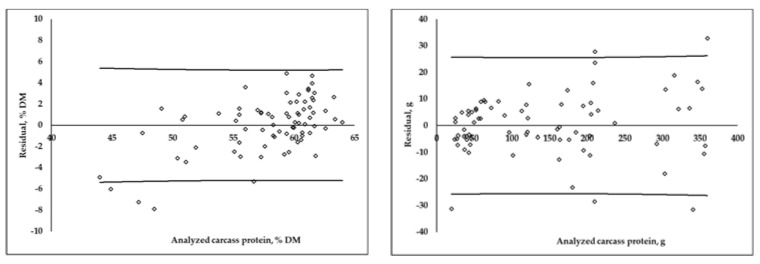
Residuals distribution (analyzed−predicted) of multiple linear regression equations for carcass protein and fat content, expressed as a DM percentage and in grams (calibration group; *n* = 75). Lines represent prediction confidence interval (95%).

**Figure 3 animals-12-02926-f003:**
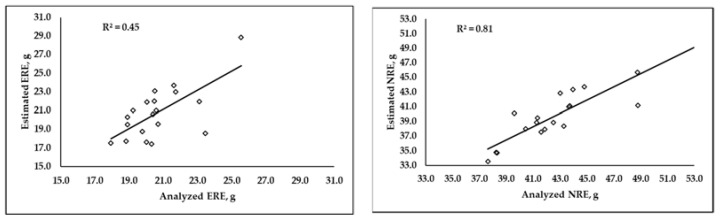
Correlation between overall Energy Retention Efficiency (ERE) and overall Digestible Nitrogen Retention Efficiency (NRE), calculated using analyzed or predicted by MLR carcass composition values.

**Table 1 animals-12-02926-t001:** Ingredients and chemical composition of lactating (Cunilactal Super (CLS)) and standard-growing (CON) feeds.

Diet	CLS	CON
Ingredient, % as-fed basis		
Wheat bran	30	30
Barley	16.6	24.3
Sunflower meal	-	11.7
Alfalfa	29.2	20
Cereal straw	2	10
Soybean oil	0.68	1
Soybean meal 47	0.77	-
Molasses	4	-
Whole-grain sunflower	15	-
L-Threonine	-	0.11
L-Lysine	0.18	0
Sodium Chloride	0.3	0.6
Monocalcium phosphate	0.41	-
Calcium carbonate	-	1.5
Sepiolite	-	0.25
Mineral and Vitamin premix A ^1^	0.86	-
Mineral and Vitamin premix B ^2^	-	0.54
Analyzed chemical composition, %		
DM	89.7	91.2
CP	17.1	15.0
NDF	32.0	36.3
ADF	17.6	19.8
ADL	4.47	4.74
Starch	16.9	16.9
Fat	3.30	3.30
Ash	7.31	7.21
Digestible energy, Kcal/kg ^3^	2450	2354
Digestible nitrogen ^3^	1.84	1.74

^1^ Provided by Trouw Nutrition España (Tres Cantos, Spain). Mineral and vitamin composition (per kg of complete diet): 240 mg of S; 240 mg of Mg as MgO; 20 mg of Mn as MnO; 75 mg of Zn as ZnO; 180 mg of Cu as CuSO_4_.5H_2_O; 1.1 mg of I as KI; 0.5 mg of Co as CoCO_3_.H_2_O; 0.06 mg of Se as SeO_2_; 7.8 mg of Fe as FeCO_3_; 12,000 UI of Vitamin A; 10,800 UI of Vitamin D3; 45 mg of Vitamin E dl-alfa-tocopherol acetate; 1.2 mg of Vitamin K; 2 mg of Vitamin B1; 60 mg of Vitamin B2; 2 mg of Vitamin B6; 10 mg of Vitamin B12; 40 mg of niacin; 20 mg of calcium pantothenate; 18.4 mg of pantothenic acid; 5 mg of folic acid; 75 mcg of biotin; 260 mg of choline chloride; 60 mg of robenidine hydrochloride (Cycostast 66 G ©); 0.12 mg of butylated hydroxyanisole; 13.2 mg of Butylated hydroxytoluene; 38.4 mg of ethoxyquin. ^2^ Provided by Trouw Nutrition España (Tres Cantos, Spain). Mineral and vitamin composition (per kg of complete diet): 240 mg of S; 240 mg of Mg as MgO; 20 mg of Mn as MnO; 75 mg of Zn as ZnO; 180 mg of Cu as CuSO_4_.5H_2_O; 1.1 mg of I as KI; 0.5 mg of Co as CoCO_3_.H_2_O; 0.06 mg of Se as SeO_2_; 7.8 mg of Fe as FeCO_3_; 12,000 UI of Vitamin A; 10,800 UI of Vitamin D3; 45 mg of Vitamin E dl-alfa-tocopherol acetate; 1.2 mg of Vitamin K; 2 mg of Vitamin B1; 60 mg of Vitamin B2; 2 mg of Vitamin B6; 10 mg of Vitamin B12; 40 mg of niacin; 20 mg of calcium pantothenate; 18.4 mg of pantothenic acid; 5 mg of folic acid; 75 mcg of biotin; 260 mg of choline chloride; 10 mg of diclazuril 0.5 g/100 g (Clinacox 0.5% Premix ©); 0.12 mg of butylated hydroxyanisole; 13.2 mg of butylated hydroxytoluene; 38.4 mg of ethoxyquin etoxiquine^3^, estimated using the digestibility coefficients of Maertens et al. [26].

**Table 2 animals-12-02926-t002:** Repeatability (S_R_) and coefficient of variation of repeatability (CV_R_) of resistance (Rs) and reactance (Xc).

	S_R_, Ω	CV_R_, %
Rs, Ω	15.3	15.9
Xc, Ω	3.44	17.6

**Table 3 animals-12-02926-t003:** Average daily feed intake (ADFI), average daily gain (ADG) and feed conversion rate (FCR) of the calibration-group animals from 35 to 77 d.

Period		ADFI, g/d	ADG g/d	FCR
35–49 d	Mean	89.1	44.0	2.02
	SD	13.6	9.86	0.222
	No.	45	45	45
49–63 d	Mean	128	41.8	3.06
	SD	20.2	11.6	1.85
	No.	30	30	30
63–77 d	Mean	147	43.0	3.41
	SD	25.0	8.50	0.836
	No.	15	15	15
35–77 d	Mean	121	42.9	2.83
	SD	19.6	5.80	0.267
	No.	15	15	15

SD: standard deviation; No.: number of animals.

**Table 4 animals-12-02926-t004:** Body weight, carcass weight (CW), carcass yield (CY) and analyzed carcass composition (expressed as a percentage and in grams) of the of the calibration-group rabbits at 25, 35, 49, 63, and 77 d of age (*n* = 15).

	Age, d					SEM	*p*-Value Linear ^1^	*p*-Value Quadratic ^2^
	25	35	49	63	77			
BW, g	370	634	1219	1912	2930	53.2	0.21	<0.001
CW, g	228	292	588	1025	1838	7.80	<0.001	<0.001
CY, %	64.3	45.7	48.1	53.6	62.8	2.81	<0.001	<0.001
Carcass chemical composition, %DM								
Water, %	71.6	72.1	70.0	67.5	64.1	0.47	0.087	<0.001
Protein	59.7	61.5	61.3	57.5	49.4	0.90	<0.001	<0.001
Fat	19.6	19.0	23.0	28.3	34.3	1.10	0.20	0.005
Ash	16.8	18.5	15.8	14.2	11.5	0.53	0.033	0.001
GE ^3^, kJ/100 g DM	2054	1980	2117	2333	2557	30.7	0.006	<0.001
Carcass chemical composition, g								
Water	164	210	411	691	1176	18.7	<0.001	<0.001
Protein	38.4	50.5	108	191	325	5.78	<0.001	<0.001
Fat	12.7	15.9	41.5	96.5	230	7.88	<0.001	<0.001
Ash	10.6	14.9	27.5	47.1	75.0	1.36	<0.001	<0.001
GE ^3^, MJ	1.32	1.63	3.78	7.84	17.0	0.40	<0.001	<0.001

^1^ Probability of linear response to animal age. ^2^ Probability of quadratic response to animal age. ^3^ GE: gross energy; kJ: kilojoules; MJ: Megajoules; DM: dry matter.

**Table 5 animals-12-02926-t005:** Comparison of the prediction equations’ accuracy assessed using an independent dataset (*n* = 25) via multiple linear regression (MLR).

	MLR		
	R^2^	MPE ^1^	RMPE ^1^, %
Chemical carcass composition, %DM			
Water	0.79	1.14	1.66
Protein	0.68	1.85	3.22
Ash	0.66	0.89	5.82
Fat	0.75	2.60	10.5
Energy, kJ/100 g DM	0.82	56.1	2.54
Chemical carcass composition, g			
Water	0.99	21.9	4.20
Protein	0.99	1.83	5.48
Ash	0.96	3.33	9.10
Fat	0.95	16.2	21.9
Energy, MJ	0.98	0.47	6.77
Carcass yield, %	0.50	5.46	10.0

^1^ MPE = mean prediction error; RMPE = relative mean prediction error.

**Table 6 animals-12-02926-t006:** Comparison between analyzed and predicted chemical composition (mean (SD)) via multiple linear regression (MLR) using a paired *t*-test.

	Analyzed	Predicted by MLR	*p*-Value
Chemical carcass composition, %DM			
Water, %	68.7 (3.32)	69.1 (3.58)	0.42
Protein	57.6 (4.96)	57.4 (5.08)	0.87
Ash	15.4 (2.38)	15.3 (2.53	0.88
Fat	24.7 (5.63)	25.5 (6.97)	0.44
Energy, kJ/100 g DM	2206 (210)	2215 (219)	0.70
Chemical carcass composition, g			
Water	523 (3.96)	520 (382)	0.74
Protein	143 (114)	139 (108)	0.26
Ash	36.6 (27.9)	34.3 (23.9)	0.08
Fat	74.1 (73.4)	82.1 (90.9)	0.24
Energy, MJ	6.16 (5.73)	6.29 (6.04)	0.49
Carcass yield, %	54.3 (6.73)	54.3 (5.47)	0.88

**Table 7 animals-12-02926-t007:** Comparison between analyzed and predicted nitrogen, fat and energy retention and Nitrogen Retention Efficiency (NRE) and Energy Retention Efficiency (ERE) in rabbits from 35 to 63 d of age (*n* = 20).

	Period			
Digestible N intake, g/d	35−49 d		1.44 (0.22)	
	49−63 d		2.57 (0.43)	
	35−63 d		1.97 (0.32)	
Digestible E intake, kcal/d	35−49 d		196 (29.8)	
	49−63 d		340 (57.5)	
	35−63 d		268 (43.6)	
Average daily gain, g/d	35−49 d		41.1 (9.21)	
	49−63 d		51.1 (10.2)	
	35−63 d		45.5 (9.70)	
Parameter	Period	Analyzed ^1^	Predicted ^2^	*p*
Nitrogen retention, g/d	35−49 d	0.654 (0.13)	0.675 (0.13)	0.30
	49−63 d	0.896 (0.15)	0.867 (0.12)	0.25
	35−63 d	0.770 (0.14)	0.771 (0.13)	0.10
Fat retention, g/d	35−49 d	2.73 (0.50)	1.93 (0.72)	0.10
	49−63 d	4.51 (0.48)	4.33 (1.66)	0.21
	35−63 d	3.62 (0.49)	3.13 (1.19)	0.15
Energy retention, kcal/d	35−49 d	35.8 (6.85)	40.1 (10.3)	0.45
	49−63 d	75.6 (9.35)	71.4 (12.9)	0.36
	35−63 d	55.7 (8.10)	55.8 (11.6	0.15
NRE, % ^3^	35−49 d	45.4 (13.9)	46.9 (11.7)	0.13
	49−63 d	35.7 (7.63)	34.5 (7.32)	0.35
	35−63 d	39.1 (4.11)	39.1 (3.23)	0.20
ERE, % ^3^	35−49 d	18.2 (5.35)	20.4 (7.29)	0.15
	49−63 d	22.2 (3.51)	21.0 (4.18)	0.22
	35−63 d	20.7 (1.85)	20.8 (2.79)	0.19

^1^ Calculated as the difference in total analyzed carcass energy and nitrogen content of animals at 35 and 63 days of age. ^2^ Calculated as the difference in total carcass energy and nitrogen content of animals by using Multiple Linear Regressions equations at 35 and 63 days of age. ^3^ ERE = Energy Retention Efficiency = 100 × (energy retained/DE Intake); NRE = Nitrogen Retention Efficiency = 100 × (N retained/DN Intake).

## Data Availability

The data are not publicy available due to intellectual property company (Trouw Nutrition) rules.

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
