# Peer review of "Application of Bioelectrical Impedance Analysis (BIA) to Assess Carcass Composition and Nutrient Retention in Rabbits from 25 to 77 Days of Age"

_animals, 2022, doi:10.3390/ani12212926_

Round 1
Reviewer 1 Report
General comments to authors
The manuscript entitled «Application of bioelectrical impedance analysis (BIA) to assess carcass composition and nutrient retention in rabbits from 25 to 77 days of age” aimed to determine and validate the prediction equations developed with BIA method to estimate in vivo carcass composition
The subject of the paper could fall within the general scope of the journal and the contribution could be original. The development of a non-invasive method, such as bioelectrical impedance, to determine the carcass composition in fattening rabbits, is very interesting not only at a scientific level but also at a commercial level and especially in an animal welfare context. Moreover, the objective is well defined. The material and methods are adequate. Therefore, I recommend its reconsideration to be publish in ANIMAL after consider the following recommendation.
Other considerations:
Abstract:
Line 24: define CP and GE
Line 33: remove (ERE)
Line 35: remove (NRE)
Introduction:
I recommend authors to slightly enlarge the introduction and divide it into paragraphs
Line 58: “poultry”
Material and methods:
No information about feed conversion rate is provided.
Table 1. review the title.
Line 126: croup?
Statistical analysis: I recommend dividing it into subsections for a better understanding
Results:
Table 2 and 3: replace mean by descriptive
Lines 206-216: (Tables 2 and 3) The difference between maximum and minimum of other characters is striking. please comment and justify
Table 6: define GE
Lines 269-279: authors should not limit to indicate whether the correlation is positive or negative and whether it is significant. For example, the correlation of reactance and fat is positive and significant but it is low (0.24). please comment.
Author Response
Thank you very much for your comments. Please, find the answer to your questions in the attached file.

Reviewer 2 Report
In my opinion the results described in the manuscript are very interesting and innovative. They have the potential to revolutionize and simplify animal research, but first, the research methods need to be improved.
However, there are some vaguenesses in this manuscript that need to be clarified. It is very useful to quickly assess the chemical composition of the meat. However, the authors evaluate the composition of the carcass together with the bones, while in typical scientific studies carried out on rabbits, the composition of individual muscles or cuts, e.g. loin, is assessed (examined). Moreover, it is incomprehensible why the authors evaluate nitrogen and energy retention for? These parameters are not entirely equivalent to the digestibility of nutrients.
The manuscript is difficult to read, it is too extensive in the methodological part concerning BIA, while the traditional research methods are described very briefly. It should be changed.
Other mistakes are listed below:
- Keywords: lack of information on N and energy retention, which accounts for a large part of the research;
- Table 1: The total component composition of CLS and CON feeds is not equal to 100%, because the sum of CLS = 100.12%, and CON = 99.96%.
Because electrodes were inserted into the body of live rabbits, the ethical aspect of the research should be well explained. The authors write that the research was approved by the Trouw Nutrition R&D Animal Care Ethical committee. It is known that Trouw Nutrition is a commercial company operating in many countries. Can such the company approve research, or is the permission of the Local Ethics Committee operating e.g. at a scientific university needed? In addition, Trouw Nutrition funded the research. However, this is a further problematic subject that I leave for the editorial decision.
The manuscript is very extensive; it should be shortened or even divided into two separate publications. I believe that it may be published, but it should also be reviewed by a specialist in the field of statistical methods and technical sciences. Only a complete evaluation of these three topics enables a proper and complete evaluation of the paper.
Author Response
Than very much you for your comments. Please, find the answer to your questions in the attached file.

Reviewer 3 Report
The Authors decided to conduct a study to validate bioelectrical impedance analysis as a tool for carcass composition assessment. This kind of study is definitely needed to provide methods for non-invasive assessment of body composition in animals. None the less, the study has few methodological flaws that I need to highlight.
First of all, the Authors performed BIA and obtained bioelectrical parameters (resistance and reactance) at living animals and then they provided the formulas on the basis of the results from bled animals. Blood contains different fractions (water, cells – protein, lipids), the amount of water also varies between the objects, so this is the main flaw of this study.
Another methodological flaw is the BIA measurement interval. The measurements were taken in long intervals, so the results are probably not comparable, as for the reliable results the animals would have to starve, not drink and not excrete throughout this time.
What is more, to assess the agreement between two methods, I suggest performing Bland-Altman analysis.
To sum up, the manuscript presents the interesting topic, however due to big methodological flaws I suggest repetition of the measurements to obtain reliable results, and then I encourage the Authors resubmit the manuscript.
Author Response

(The authors gave the same response as above.)

Reviewer 4 Report
Application of bioelectrical impedance analysis (BIA) to assess carcass composition and nutrient retention in rabbits from 25 to 77 days of age
This work applies bioelectrical impedance analysis (BIA) to estimate in vivo carcass composition and determine nutrient retention efficiencies of growing rabbits. It is a work with merit for the scientific knowledge of the body composition of rabbits.
The article is well organized and well written and, as such, is simple to follow. Accurate references support the text. The Tables and Figures are essential for understanding the article, but some corrections are necessary to improve its clarity. For example, authors should consider improving Figures 2 and 3. The material and methods are clearly described, which allows a perfect understanding of what has been done. The results are well presented and well discussed. Finally, the results corroborate the conclusions.
Some detailed comments are below:
L23-24 Please consider rephrasing “At each selected age, all the animals were stunned and bled, and the chilled carcass was analyzed for determining water, fat, CP, ash, and GE.” with “All the animals were stunned and bled at each selected age, and the chilled carcass was analyzed for determining water, fat, CP, ash, and GE.”
L55 after “(TOBEC) in growing rabbits” a reference is missing”.
L138 Please change “on carcass between” with “on carcasses between”
L139 Please change “Body weight and feed intake of those animals was registered, ” with “Body weight and feed intake of those animals were registered”
L140 Please change “rate was calculated ” with “rate were calculated ”
L154 Please change “by lineal regression ” with “by linear regression ”
L188 Please consider rephrasing “For validating the regression equations with independent data, the chemical composition data of the VG animals were used. ” with “The chemical composition data of the VG animals were used to validate the regression equations with independent data”
L237 Please change “age, therefore, ” with “age; therefore,”
L253 Please change “%DM respectively ” with “%DM, respectively”
L326 Please change “Distribution of residuals obtained by differences, between ” with “The distribution of residuals obtained by differences between”
L331 Please change “On the contrary when ” with “On the contrary, when”
L38-389 Please change “That is the reason why in rabbits it´s recommended, to take at least two measurements per animal ” with “That is why it is recommended for rabbits to take at least two measurements per animal.”
L392 Please change “changes on the chemical ” with “changes in the chemical ”
L399 Please change “deposited on carcass, ” with “deposited on the carcass,”
L401 Please change “growth rate of animal’s tissues ” with “growth rates of animal tissues”
L404 Please consider rephrasing “Since this moment, the rabbit starts to deposit more fat and the carcass yield increases with age, as observed in previous studies [44]. ” with “Since then, the rabbit starts depositing more fat and the carcass yield increases with age, as observed in previous studies [44].”
L435 Please change “Besides it is ” with “Besides, it is ”
L463 Please change “Partridge et al., (1989) ” with “Partridge et al. (1989) ”
L466 Please change “De Blas et al. (1985) [5], observed a retention ” with “De Blas et al. (1985) [5] observed a retention”
Author Response
Thank you very much for your comments. Please, find the answer to your questions in the attached file

Round 2
Reviewer 2 Report
I have no comments
Author Response
Thank you very much.
Reviewer 3 Report
The Authors answered reasonably to my comments, however I still have few comments.
C1: You used abbreviations in the abstract, please explain them the first time they are used.
C2: Generally, if P>.01 then the P value should be expressed to 2 digits whether or not it is significant. When rounding, 3 digits is acceptable if rounding would change the significance of a value (eg, you may write P=.049 instead of .05). If P<.01, it should be expressed to 3 digits. Please correct it.
C3: Regarding your response: “Another methodological flaw is the BIA measurement interval. The measurements were taken in long intervals, so the results are probably not comparable, as for the reliable results the animals would have to starve, not drink and not excrete throughout this time.
You are right, and this variability is in the error term. It was done in this way because the final aim is that this method can be applied at field level. In standard conditions when we take measurements, we cannot know weather an animal has drunk, eaten excreted or not. We tried to take the BIA measurements without modifying the natural behavior of the animals.”
- Please include this information in the limitations of the study section.
C4: In the figure 2, there are limits of the agreement lines missing. Please add them and on the basis of that provide B-A index value (the percentage of the results classified out of the 95% limit of agreement among all the measurements) to prove the validation of the method (it should not exceed 5% to provide full validation of the method).
Author Response
Dear reviewer, please find enclosed the responses to your comments in the attached file. Thank you very much.

Round 3
Reviewer 3 Report
Q1: Figure 1 – The quality of the figure is unacceptable. It seems to be MS Word print screen as there are red undermarks. Please correct it.
Q2: Figure 2 – Please provide index value (the percentage of the results classified out of the 95% limit of agreement among all the measurements) to prove the validation of the method (it should not exceed 5% to provide full validation of the method).
Q3: “C2: Generally, if P>.01 then the P value should be expressed to 2 digits whether or not it is significant. When rounding, 3 digits is acceptable if rounding would change the significance of a value (eg, you may write P=.049 instead of .05). If P<.01, it should be expressed to 3 digits. Please correct it.
It has been corrected (L261 and L263).”
– there are needed analogical corrections in the rest of the manuscript e.g. Table 4, paragraph 3.3 and more. Please correct them.
Author Response
Thank you very much for your review. In the attached file you can find the answers to your comments.
